# Spurious Privacy Leakage in Neural Networks

**Chenxiang Zhang**  *chenxiang.zhang@uni.lu*
*Department of Computer Science*
*University of Luxembourg*

**Jun Pang**  *jun.pang@uni.lu*
*Department of Computer Science*
*University of Luxembourg*

**Sjouke Mauw**  *sjouke.mauw@uni.lu*
*Department of Computer Science*
*University of Luxembourg*

**Reviewed on OpenReview:** *https: // openreview. net/ forum? id= tRXDCIgvTT*

## Abstract

Neural networks trained on real-world data often exhibit biases while simultaneously being vulnerable to privacy attacks aimed at extracting sensitive information. Despite extensive research on each problem individually, their intersection remains poorly understood. In this work, we investigate the privacy impact of spurious correlation bias. We introduce *spurious privacy leakage*, a phenomenon in which spurious groups are significantly more vulnerable to privacy attacks than non-spurious groups. We observe that privacy disparity between groups increases in tasks with simpler objectives (e.g. fewer classes) due to spurious features. Counterintuitively, we demonstrate that spurious robust methods, designed to reduce spurious bias, fail to mitigate privacy disparity. Our analysis reveals that this occurs because robust methods can reduce reliance on spurious features for prediction, but do not prevent their memorization during training. Finally, we systematically compare the privacy of different model architectures trained with spurious data, demonstrating that, contrary to previous work, architectural choice can affect privacy evaluation.

## 1 Introduction

Neural networks are applied across diverse domains such as face recognition, medical prognosis, or personalized advertisement. All these applications are built using user-sensitive data that can be of interest to attackers. Additionally, real-world collected data in specific domains can be limited and biased towards specific groups, a subset of the dataset sharing a common characteristic (e.g. gender, ethnicity, or geographic location). On top of the privacy concerns, neural networks trained on real-world data can inherit different kind of biases, causing failures at test time (Sagawa et al., 2019; Geirhos et al., 2020; Shah et al., 2020). These observations suggest that models trained and deployed in sensitive domains should satisfy multiple constraints, such as ensuring fair performance across groups and protection of sensitive data.

In this work, we examine the privacy consequences of *spurious correlation* bias, a statistical relationship between two variables that appears to be causal but is either caused by a third confounding variable or randomness. Such correlations are prevalent in real-world datasets, particularly in domains with limited and imbalanced group representation. To address the problem, several mitigation methods have been proposed in the literature with the objective to improve the performance of the worst-group (Sagawa et al., 2019; Nam et al., 2020; Liu et al., 2021a; Izmailov et al., 2022; Yang et al., 2023). However, their privacy implications have not been studied. On the other hand, privacy research typically focuses on simple datasets, such as CIFAR (Krizhevsky, 2009; Hu et al., 2022), overlooking the privacy risks associated with realistic applications

that often use biased datasets. These limitations can lead to additional concerns. From an ethical view, an unequal privacy treatment exacerbates existing inequalities, particularly when spurious correlations are sensitive attributes (e.g. race or gender). From a technical perspective, privacy research often assumes that attacks equally affect all samples in the training data. But a spurious correlated dataset can be represented as subgroups with different properties. This raises a natural question: is there a significant privacy risk difference between subgroups? For example, consider a medical dataset where 'young/healthy' and 'old/sick' dominate, with a few 'young/sick' cases. A model can memorize this uncommon (spurious) correlation, at the risk of exposing the 'young/sick' patients to further privacy vulnerability. This inequality adds on top of the fairness problem: groups already suffering from poor performance can simultaneously face higher privacy risks, violating both performance and privacy equity principles. In such a scenario, how can an auditor detect these fairness and privacy issues related to data biases?

Motivated by these observations, we investigate privacy with spurious correlations to understand how real-world dataset biases can affect privacy vulnerabilities. In practice, we analyze the privacy of deep neural network models trained on real-world spurious correlated datasets using *membership inference attacks* (MIA), a family of privacy attacks commonly used for their simplicity and versatility (Murakonda & Shokri, 2020; Carlini et al., 2021).

**Contributions.** We identify a phenomenon we term *spurious privacy leakage*, where groups with spurious correlation are significantly more vulnerable to MIA than other groups, creating a privacy disparity (Section 3.1). This phenomenon adds a fairness requirement on top to the existing privacy challenges, modeling a realistic scenario for data-sensitive applications. Prior works demonstrated how privacy auditing may naively conclude that a model satisfies the privacy requirements by evaluating on an aggregated average metric over the entire dataset (Carlini et al., 2022; Ye et al., 2022; Sablayrolles et al., 2019). We demonstrate that this also applies to realistic spurious data, where an average metric evaluation is misleading for evaluating spurious and non-spurious groups. Addressing this privacy disparity is important to bridge and advance the privacy and spurious correlation research, to characterize the risks of the model, and to improve the auditing process.

We continue with further analysis on *spurious privacy leakage.* Prior works suggested that improving fairness can, in limited scenarios, bound privacy disparity (Kulynych et al., 2022). Given the range of methods proposed to mitigate spurious correlations (equivalent as improving fairness between different data groups) (Sagawa et al., 2019; Kirichenko et al., 2022; Liu et al., 2021a; Zhang et al., 2022), we investigate whether these mitigations can also help to reduce privacy leakage. To explore this, we apply robust training methods to mitigate spurious correlations and re-evaluate the privacy vulnerability. Surprisingly, we find no consistent and significant privacy improvement (Section 4.1). This observation leads us to introduce a new perspective on group privacy disparity based on memorization (Zhang et al., 2021): spurious robust training improves worst-class performance but does not reduce the memorization level of data compared to standard training (Figure 8), and thus neither can mitigate the privacy risks. Additional results confirm that robust training essentially learns the same features as standard training. Moreover, we also assess the effectiveness of differential privacy to mitigate the spurious privacy leakage. We observe that given a pretrained model, a stricter privacy guarantee can offer worst group privacy protection at the expense of reduced utility (Section 4.2).

Lastly, we analyze the influence of model architecture on privacy disparity. While prior works mostly focus on ResNet-like architecture (Carlini et al., 2022; Liu et al., 2022a), we fairly compare the performance and privacy of different model families. Our evaluation include state-of-the-art convolutional and transformer-based architectures, pretrained using both supervised and self-supervised training. Contrary to prior works showing that the best shadow architecture always matches the target one (Carlini et al., 2022; Liu et al., 2022a), our results provide counterexamples when models are trained on spurious correlated data. This highlights a key difference in privacy behavior when models are trained on biased real-world data. We release the code at https://github.com/orientino/spurious-mia.

## 2 Background & Related work

We provide a concise introduction to key topics necessary to follow the rest of this work, including neural networks, membership inference attacks, and spurious correlation.

**Neural networks** represent functions $f_{\boldsymbol{\theta}} \colon \mathcal{X} \to \mathcal{Y}$ that map the input data $\boldsymbol{x} \in \mathcal{X}$ to a label $\boldsymbol{y} \in \mathcal{Y}$. The dataset $\mathcal{D} = \{(\boldsymbol{x}_i, \boldsymbol{y}_i)\}$ is a set of labeled pairs used for estimating the model parameters. The neural network is parametrized by $\boldsymbol{\theta} \in \mathbb{R}^n$ and it is updated using a first-order optimizer to minimize a loss function $\ell \colon \mathcal{Y} \times \mathcal{Y} \to \mathbb{R}$. We focus on the classification setting where the cross-entropy loss is commonly used. Formally, the objective is the *empirical risk minimization* (ERM) (Vapnik, 1991):

$$\hat{\boldsymbol{\theta}}_{\text{ERM}} = arg \min_{\boldsymbol{\theta}} \mathbb{E}_{(\boldsymbol{x}, \boldsymbol{y}) \in \mathcal{D}}(\ell(\boldsymbol{y}, f_{\boldsymbol{\theta}}(\boldsymbol{x})))$$

**Spurious correlation** is a statistical relationship between two variables $X$ and $Y$ that first appears to be causal but in reality is either caused by a third confounding (e.g. spurious) variable $Z$ or due to random chance. This relationship is in contrast with causality, where the change of the variable $X$ leads to a direct and predictable outcome of $Y$ while ruling out the presence of any confounding factors $Z$. For a given dataset with spurious correlation, a feature $\boldsymbol{z}$ is called spurious if it is correlated with the target label $\boldsymbol{y}$ in the training data but not in the test data. For example, in a binary bird classification dataset where waterbirds mainly appear on a water background, a biased model can exploit the background spurious feature instead of the bird invariant feature, leading to a wrong prediction when the input is a waterbird on a land background (Sagawa et al., 2019). Ideally, we would like to suppress the bias coming from the spurious features, which can be expressed as $\Pr(\boldsymbol{y} \mid \boldsymbol{x}) = \Pr(\boldsymbol{y} \mid \boldsymbol{x}_{\text{inv}}, \boldsymbol{z}) = \Pr(\boldsymbol{y} \mid \boldsymbol{x}_{\text{inv}})$ where we decomposed the input $\boldsymbol{x}$ as a combination of invariant features $\boldsymbol{x}_{\text{inv}}$ and spurious features $\boldsymbol{z}$. Sagawa et al. (2019) proposed the group *distributionally robust optimization* (DRO) to mitigate spurious features. DRO minimizes the worst-group loss, differing from ERM which minimizes the average loss. Formally, the objective function of DRO is defined as:

$$\hat{\boldsymbol{\theta}}_{\text{DRO}} = arg \min_{\boldsymbol{\theta}} \max_{\boldsymbol{g} \in \mathcal{G}} \mathbb{E}_{(\boldsymbol{x}, \boldsymbol{y}, \boldsymbol{g}) \in \mathcal{D}}[\ell(\boldsymbol{y}, f_{\boldsymbol{\theta}}(\boldsymbol{x}))]$$

where the dataset is divided into $|\mathcal{G}|$ groups. The new dataset is $\mathcal{D} = \{(\boldsymbol{x}_i, \boldsymbol{y}_i, \boldsymbol{g}_i)\}$ where $\boldsymbol{g} \in \mathcal{G}$ is a discrete-valued label (e.g. all the combinations of birds and backgrounds or geographical area information (Koh et al., 2021)). DRO is considered an oracle method due to its explicit use of the group information for the training (Liu et al., 2021a). Additional methods in the literature suppress the spurious features by learning and assigning a different weight per sample (Liu et al., 2021a; Nam et al., 2020), by retraining the classifier head at the end of the training (Kirichenko et al., 2022; Izmailov et al., 2022; Kang et al., 2020), by group sampling (Yang et al., 2024; Idrissi et al., 2022), or using contrastive methods (Zhang et al., 2022).

**Membership inference attacks** (MIA) aim to determine whether a specific input data was used during the model training. MIA is usually used to audit a model's privacy level thanks to its simplicity (Murakonda & Shokri, 2020) and versatility for creating a more complex attack (Carlini et al., 2021). The membership inference problem can be defined as learning a function $\mathcal{A} \colon \mathcal{X} \to [0, 1]$, where $\mathcal{A}$ is the attacker model that takes input $\boldsymbol{x} \in \mathcal{X}$ and outputs 1 if $\boldsymbol{x}$ was used during the model training. We assume the black-box (Shokri et al., 2017) access to the target model, where the only target information accessible is the output probability vector $\boldsymbol{p}$. Shokri et al. (2017) introduced the first MIA for neural networks with black-box access, where several *shadow* models are trained to mimic the behavior of the *target* model. More advanced attacks have been developed based on the idea of shadow models (Yeom et al., 2018; Liu et al., 2022a; Carlini et al., 2022; Ye et al., 2022; Sablayrolles et al., 2019; Watson et al., 2022; Long et al., 2020). In this work, we focus on the state-of-the-art LiRA method (Carlini et al., 2022). Given an input $\boldsymbol{x}$, LiRA predicts its membership by training $N$ shadow models, each on a different subset of the dataset. Half of the models are named INs and contain $\boldsymbol{x}$ and the other half named OUTs do not. Each shadow model IN outputs a confidence score $\phi(\boldsymbol{p}_{\text{shadow}})$ which is used to estimate the parameters of a Gaussian $\mathcal{N}(\mu_{\text{in}}, \sigma_{\text{in}})$, and in the same way, OUTs are used to estimate $\mathcal{N}(\mu_{\text{out}}, \sigma_{\text{out}})$. Finally, the result of the attack is defined as a likelihood-ratio test:

$$\Lambda = \frac{\Pr(\phi(\boldsymbol{p}_{\text{target}}) \mid \mathcal{N}(\mu_{\text{in}}, \sigma_{\text{in}}))}{\Pr(\phi(\boldsymbol{p}_{\text{target}}) \mid \mathcal{N}(\mu_{\text{out}}, \sigma_{\text{out}}))}$$

where $\boldsymbol{p}$ is the model's output as a probability vector, $\phi(\boldsymbol{p}_{\text{target}}) = log(\boldsymbol{p}/(1 - \boldsymbol{p}))$ is the confidence score obtained by querying the target model with $\boldsymbol{x}$. The score $\Lambda$ is used by the attacker to determine how likely it is that the given $\boldsymbol{x}$ is a member.

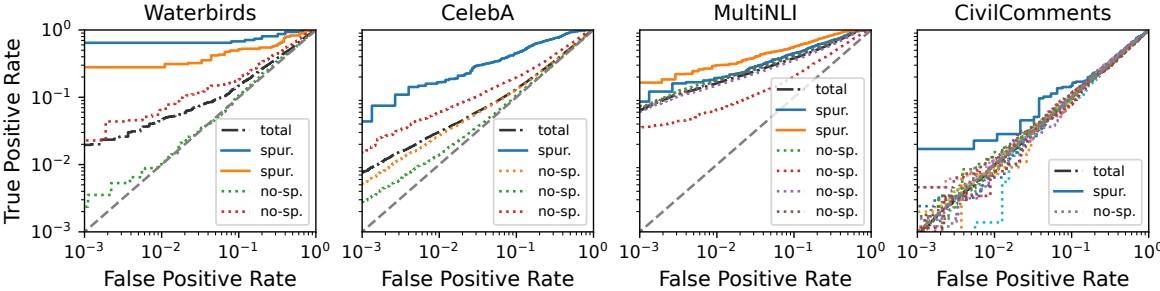

Figure 1: Attack success rate divided per group on Waterbirds, CelebA, MultiNLI, and CivilComments respectively. Across the datasets, there is a spurious group (solid lines) with consistent higher privacy leakage compared to non-spurious groups under the LiRA attack.

**Privacy and safety.** The intersection of privacy and ML safety topics has been extensively studied. Wang et al. (2021) focused on how pruning can mitigate privacy attacks, Shokri et al. (2021) explored the connection between privacy and explainability, Song et al. (2019) found that adversarial training can increase privacy leakage, but however, Li et al. (2024) reported contradictory findings when using a better privacy evaluation (Carlini et al., 2022). In our work, we investigate the privacy risk of real-world spurious correlated datasets, which is related to fairness where the goal is to ensure equal performance across groups. Prior works on privacy-fairness reported that subpopulations can exhibit varying levels of privacy risk (Tian et al., 2024; Zhong et al., 2022; Kulynych et al., 2022). In particular, Tian et al. (2024) showed that fairness methods can mildly mitigate MIA risks using the *average* metrics. Instead, we use a *per-group* analysis, revealing the spurious privacy leakage phenomenon with group privacy disparity between spurious and non-spurious data. Kulynych et al. (2022) and Zhong et al. (2022) also investigated privacy disparity on synthetic and tabular datasets. They hypothesize that, under certain conditions, group fairness improvements or differential privacy can be effective as privacy mitigation. In Section 4.1, we show that neither approaches can effectively address privacy disparities in real-world datasets. Similar to our setting, Yang et al. (2022) studied the privacy risks associated with a toy dataset (MNIST) using a suboptimal evaluation. In contrast, we use state-of-the-art methods to evaluate real-world datasets. Lastly, while out-of-distribution data are known to be more vulnerable to privacy attacks than in-distribution data (Carlini et al., 2022), spurious correlated data differ from out-of-distribution data, as they are by definition in-distribution. Our contributions present a set of results to resolve the conflicts in the literature while also offering new directions for future works.

## 3 Privacy risks of spurious correlated data

We investigate the privacy leakage of real-world spurious correlated data. Our results show that auditing the privacy level on the whole dataset is misleading when spurious correlations are present, as spurious groups have significantly and consistently higher privacy leakage compared to non-spurious groups. Moreover, we find that *spurious privacy leakage* is more severe when models are trained on simpler tasks.

### 3.1 Spurious privacy leakage

Spurious correlations are characterized by the presence of spurious features. Assuming we have the labels of the spurious features, learning with spurious correlation is equivalent to learning with an imbalanced dataset. We refer to spurious groups as the minority groups with the worst performance (e.g. worst-group accuracy) compared to the majority groups.

*Experiment setup.* We select the datasets that are used by the spurious correlation community (Yang et al., 2023): Waterbirds (Sagawa et al., 2019), CelebA (Liu et al., 2015), FMoW (Koh et al., 2021), MultiNLI (Williams et al., 2017), and CivilComments (Koh et al., 2021). These datasets contain real-world spurious correlations, diverse modalities, and different target complexity (see Appendix A for details). Moreover, to the best of our knowledge, we are the first to study MIA attacks on subgroups of these realistic datasets. We use

the pretrained ResNet50 (He et al., 2016) on ImageNet1k and finetune using random crop and horizontal flip. For text datasets, BERT's bert-base-uncased model (Devlin et al., 2019) is used. We perform hyperparameter optimization for each dataset using a grid search over learning rate (lr), weight decay (wd), and epochs. The grid search and its best hyperparameters are in Appendix B. We report the training and test accuracy to evaluate the performance and their difference to quantify the overfitting level. We do the same for the worst-group accuracy (WGA), which is a commonly used proxy metric to measure the mitigation success of spurious features (Sagawa et al., 2019). For privacy evaluation, we follow the guidelines from Carlini et al. (2022). We train non-overfit models and report the full log-scale ROC curves, the true positive rate (TPR) at a low false positive rate (FPR) region, and also the AUROC curve for completeness. We train 32 shadow models for Waterbirds/CelebA and 16 for FMoW/MultiNLI.

Across all the spurious correlated datasets, the group performance disparity is consistently present, with the spurious groups having the lowest accuracy among all the groups (see Table 5). For example, in Waterbirds, ERM has a test *average* accuracy of 81.08% while one of the spurious group only 34.41%. Beyond these performance disparity, we now show that spurious correlations also cause privacy issues. Using the state-of-the-art MIA method LiRA (Carlini et al., 2022), we analyze the privacy leakage of each group of the five spurious correlated datasets. For each dataset, we train the shadow models using 50% of the sampled training data as in the LiRA algorithm. We ensure that the sampled subset maintains a similar group proportion as the original dataset by first sampling per group, and then combining all the sampled groups together. Our results in Figure 1 show that across the datasets, there exists a spurious group that exhibits higher privacy leakage than non-spurious groups. The largest disparity is observed at ≈3% FPR area of Waterbirds, where the samples in the most *spurious group are* ≈10 *times more vulnerable than samples in the non-spurious group.* In CelebA, we continue to observe a significant disparity, with the most spurious group being ≈10 times more vulnerable than the least spurious group. In both the text datasets MultiNLI and CivilComments, the disparity is milder with ≈4 times difference between the most and least vulnerable groups (see Table 1 for the exact TPR at low FPR). The existence of spurious privacy leakage unfairly exposes some data groups, allowing an attacker to craft better targeted attacks. Our results on spurious correlation extend prior related research on fairness (Zhong et al., 2022; Kulynych et al., 2022; Tian et al., 2024), where they also found privacy disparity between different subpopulations. Surprisingly, we do not observe a spurious privacy leakage in the FMoW dataset, which we investigate in the next section.

> **Finding I.** *Spurious privacy leakage is present in real-world datasets, where spurious groups can have disproportionately higher vulnerability to membership inference attacks than other groups.*

## 3.2 Connecting privacy leakage with task complexity

The FMoW dataset exhibits a similar privacy vulnerabilities across the all the groups. FMoW requires to classify 62 classes compared to the simpler binary classification task of Waterbirds or CelebA. Given a fixed dataset with spurious correlation, we show the relationship between the number of classes, the complexity of learned features, and *spurious privacy leakage*. In particular, a simpler task induces to a simpler feature representation and a higher group privacy disparity.

*Experiment setup.* We create two new datasets with 16 (FMoW16) and 4 (FMoW4) classes by sequentially clustering the 62 classes of the original FMoW. We train 16 shadow models for each dataset as in Section 3.1 and use LiRA for privacy analysis. The results are averaged over 5 different target models.

Figure 2a shows the decrease of average privacy risk over the total dataset as the task simplifies from 62 to 4 classes (black dot-dashed line). This observation is consistent with prior works on balanced datasets. For example, CIFAR100 has a higher privacy leakage compared to its counterpart CIFAR10 (Shokri et al., 2017; Carlini et al., 2022). The same is found in segmentation tasks where a larger output dimension has higher MIA vulnerability (Shafran et al., 2021). These findings confirm the importance of task complexity on privacy risks. We continue the analysis by investigating at a *per-group* level on top of the *average* one. Interestingly, we observe that *the privacy disparity between the spurious and non-spurious groups grows as the task complexity simplifies.* While the leakage for most of the groups drops, the spurious group 2 remains consistently vulnerable at 6% TPR at 0.1% FPR across the dataset with different classes.

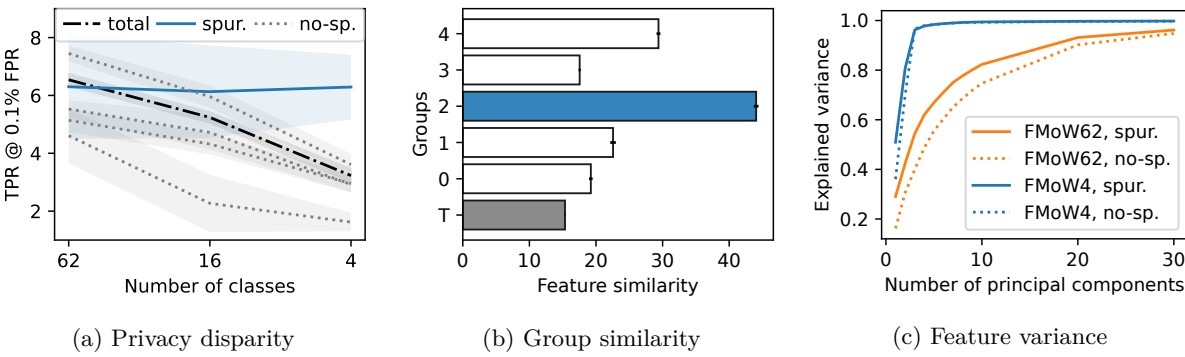

(a) Privacy disparity        (b) Group similarity        (c) Feature variance

Figure 2: (a) Group privacy disparity increases as the target complexity reduces from FMoW62 to FMoW4. The solid line, representing the spurious group 2, remains constant while the other groups become less vulnerable. (b) Feature similarity of each group between FMoW62 and FMoW4 using linear CKA. The most similar group is the spurious group 2 colored in blue. (c) Feature complexity using explainable variance for embeddings of models trained on FMoW62 and FMoW4. FMoW4 requires only 3 principal components compared to 25 of FMoW62 to explain $\approx 95\%$ of the variance. Additionally, spurious groups need fewer components than non-spurious groups.

We hypothesize that spurious groups are characterized by *similar* and *fewer* discriminative features across tasks of different complexity (e.g. FMoW62 vs FMoW4), contributing to spurious privacy leakage. To quantify the feature *similarity*, we compute linear centered kernel alignment (CKA; Kornblith et al. (2019)) between feature embeddings of two models. For feature *complexity*, we apply principal component analysis (PCA) and measure the number of components required to explain a variance threshold $\tau$. Formal definitions are provided in Appendix B.1. First, we find that CKA scores between spurious group embeddings across FMoW62 and FMoW4 are significantly higher than those for non-spurious groups, indicating more similar feature representations (Figure 2b; blue bar). Second, PCA analysis show that models trained on the simpler FMoW4 task require far fewer components to reach $\tau = 0.95$ compared to FMoW62 (3 vs 25 components), indicating reduced feature complexity (Figure 2c). Within each task, spurious groups also require fewer components than non-spurious groups. These results suggest that task simplification leads to increased feature alignment in spurious groups and reduced feature complexity, which exacerbate privacy disparity. In contrast, more complex tasks lead to richer representations but also increase the average privacy vulnerability across groups.

> **Finding II.** *Spurious privacy leakage emerges as the task complexity decreases.*

## 4 Privacy risks of robust methods

Several methods have been proposed to mitigate spurious correlations (see Section 2). Counterintuitively, we find that mitigating the impact of spurious features does not reduce *spurious privacy leakage*. Although these methods improve utility fairness, group privacy disparity persists due to *memorization* (Zhang et al., 2021). Moreover, we demonstrate the practical limitations of differential privacy with spurious correlated data.

### 4.1 Privacy risks of spurious robust methods

Spurious correlations can be suppressed using training methods such as group *distributional robust optimization* (DRO) (Sagawa et al., 2019) or *deep feature reweighting* (DFR) (Kirichenko et al., 2022). Extensive benchmarks (Izmailov et al., 2022; Yang et al., 2023) report DRO and DFR as the most effective approaches for mitigating spurious correlations. DRO is referred as an oracle method because it requires a group label to minimize the worst-group error in its objective function (Liu et al., 2021a). While DFR empirically achieves the highest average worst-group accuracy compared to 17 spurious robust methods across 12 datasets (Yang et al., 2023).

Table 1: Comparing the privacy leakage of spurious robust methods per group. Although these methods improve the worst-group accuracy, DRO and DFR do not consistently mitigate the attack across datasets. *Waterbirds is evaluated at ≈3% FPR due to the limited samples in the spurious groups (see Table 6 for the complete results). Bolded values represent the best training method for privacy mitigation. The spurious groups are highlighted.

| | **TPR @ 0.1% FPR (↓)** | | |
|---|---|---|---|
| **Data** | **ERM** | **DRO** | **DFR** |
| Waterb.* | **3.12 ± 0.10** | 3.12 ± 0.11 | 3.13 ± 0.10 |
| | **30.91 ± 2.81** | 31.06 ± 2.76 | 33.20 ± 2.83 |
| CelebA | 0.27 ± 0.01 | **0.26 ± 0.01** | **0.26 ± 0.01** |
| | 4.61 ± 0.50 | **4.56 ± 0.48** | 4.77 ± 0.46 |
| MultiNLI | 5.88 ± 0.42 | **5.73 ± 0.45** | 5.86 ± 0.40 |
| | 8.26 ± 0.55 | 9.08 ± 1.37 | **7.78 ± 0.57** |
| CivilCom. | **0.10 ± 0.02** | 0.14 ± 0.03 | 0.12 ± 0.03 |
| | 0.44 ± 0.10 | 0.43 ± 0.17 | **0.32 ± 0.12** |
| FMoW | **7.45 ± 0.27** | - | 7.61 ± 0.28 |
| | **6.30 ± 1.80** | - | 6.42 ± 1.80 |

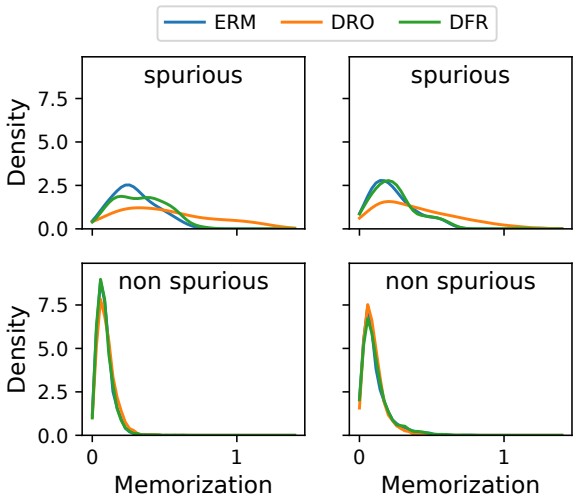

Figure 3: Memorization score per group for each spurious robust method on Waterbirds. Neither DRO nor DFR training can effectively mitigate data memorization.

We choose on these two methods in our analysis, comparing the privacy leakage of models trained with ERM, DRO, and DFR.

*Experiment setup.* For each dataset and training method, we train the shadow models by following the same LiRA setup as in Section 3. We ensure that models across different training methods use the same subset of data by fixing the random seeds. For privacy evaluation, we train non-overfit models by monitoring the difference between the train-val losses (Yeom et al., 2018).

Prior work by Yeom et al. (2018) showed that overfitting is a sufficient condition for MIA to succeed. Motivated by this, we apply robust training to reduce overfitting across groups and reassess privacy vulnerability. Kulynych et al. (2022) suggested that improving group utility fairness can mitigate privacy disparity, but only under limited conditions. More recently, Tian et al. (2024) examined the privacy implications of fairness methods over the entire dataset, without taking into account group privacy disparities. To the best of our knowledge, no prior work has analyzed the privacy implications of spurious robust methods to ensure group fairness. This raises the question: *Does mitigating performance disparity also reduce spurious privacy leakage?*

First, we apply spurious robust training methods to reduce overfitting of the spurious groups (see Appendix Table 5). As expected, DRO and DFR significantly reduce the difference between train-test WGA by mitigating the influence of spurious features on all five datasets (except for DRO on FMoW). We then run LiRA using ERM trained shadow models and use ERM, DRO, and DFR models as targets. Table 1 reports the privacy attack success rate for each dataset, group, and robust training method. We observe that, across all datasets and for both spurious and non-spurious groups, the privacy vulnerability remain almost unchanged for ERM, DRO, and DFR. This indicates that *spurious privacy leakage* persists despite the mitigation of spurious correlations. While this result may seem surprising, overfitting is not a necessary condition for MIA to succeed (Yeom et al., 2018), which explains the ineffectiveness of spurious robust training in reducing privacy disparity. We next present an alternative perspective on privacy disparity through the lens of memorization.

**Memorization of spurious robust methods.** We analyze *spurious privacy leakage* through the lens of data memorization. Memorization occurs when a model captures instance-specific features rather than learning generalizable patterns (Feldman, 2020; Feldman & Zhang, 2020). This phenomenon is tied to the success of MIA, especially for LiRA, which estimates membership by approximating the degree of memorization (Carlini et al., 2022) (see Appendix B.3).

As shown in Figure 3, spurious groups exhibit higher memorization scores and therefore are also more vulnerable to LiRA than non-spurious groups. Importantly, spurious robust training fails to reduce this memorization. The methods ERM and DFR have nearly identical memorization score distributions across both spurious and non-spurious groups. Since DFR re-trains only the final layer, it cannot address memorization which is a phenomenon distributed across multiple layers of the network, as demonstrated by Maini et al. (2023). Therefore, DFR does not address spurious privacy leakage either. DRO shows a similar memorization level as ERM for non-spurious groups, but exhibits even higher memorization for spurious groups. Using the CKA similarity metric, we investigate the embedding similarity of models trained with ERM and DRO. We find that the learned embeddings have highly similar feature representations (Section 3.2). In particular, Figure 4 shows that the CKA scores between embeddings from ERM and DRO are comparable to those between models trained with the same spurious method (ERM–ERM or DRO–DRO). This aligns with findings from Izmailov et al. (2022), who showed that applying DFR on top of a DRO trained model offers no additional utility gains.

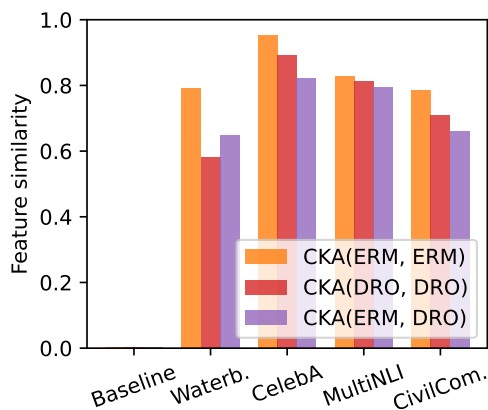

Figure 4: Feature similarity of different training objectives using linear-CKA. For the baseline, we have near-zero similarity between models trained on different datasets (Waterbirds and CelebA). A pair of models trained with different training methods (ERM, DRO) on the same dataset exhibits similar similarity as those trained using the same method, (ERM, ERM) or (DRO, DRO). Results are averaged over 5 random seeds.

> **Finding III.** *Spurious robust methods can reduce group performance disparity but fail to address spurious privacy leakage, as data memorization is largely unaffected.*

## 4.2 Differential privacy for spurious correlations

Differential Privacy (DP) provides formal guarantees against membership inference attacks (Dwork, 2006) (Definition D.1). For neural networks, DP-SGD (Abadi et al., 2016) enforces these guarantees by modifying the standard SGD optimizer. We audit models trained with DP using a practical evaluation, LiRA.

*Experiment setup.* We train target models using `opacus` (Yousefpour et al., 2021) following the guidelines from De et al. (2022). We use a large batch size of 1024 for Waterbirds and CelebA. The model selection uses a grid search with lr in [1, 1e-1, 1e-2, 1e-3], privacy guarantee $\epsilon$ in [1, 2, 8, 32, 128], and failure probability $\delta$ = 1e-4 for Waterbirds and $\delta$ = 1e-5 for CelebA. As in De et al. (2022); Panda et al. (2024), we do not use weight decay and the cosine scheduler due to optimization instability. Each model is trained up to 100 epochs for both Waterbirds and CelebA by selecting the best validation WGA checkpoint. For CelebA, we use a ConvNeXt-T (Liu et al., 2022b) pretrained on ImageNet-1k. For Waterbirds, we train from scratch due to dataset overlap with ImageNet-1k, which would invalidate privacy guarantees.[1] The privacy attack uses LiRA with 32 ERM shadow models and we report the privacy budget as $(\epsilon, \delta)$.

Figure 5 (left) presents the results of large batch trained target models across four metrics, the average and worst-group utility and privacy metrics as a function of the privacy budget $\epsilon$. Prior works by Bagdasaryan et al. (2019) shows that DP training with small batch size increases the group utility disparity, harming the worst-group utility. However, consistent with Panda et al. (2024), we observe that DP training with large batch size can help to mitigate this disparity. For example, on CelebA, a tight budget of $\epsilon = 1$ has the lowest accuracy, but a relaxed $\epsilon$ improves the worst-group accuracy. Note that this trend does not hold for models

---

[1]The Waterbirds dataset combines Caltech-UCSD Birds and Places datasets; the former overlaps with ImageNet-1k (Tramèr et al., 2024) (see https://www.vision.caltech.edu/datasets/cub_200_2011/). To the best of our knowledge, CelebA has no such overlap.

trained from scratch on Waterbirds, which lacks a "good" weight initialization. In the spurious correlation

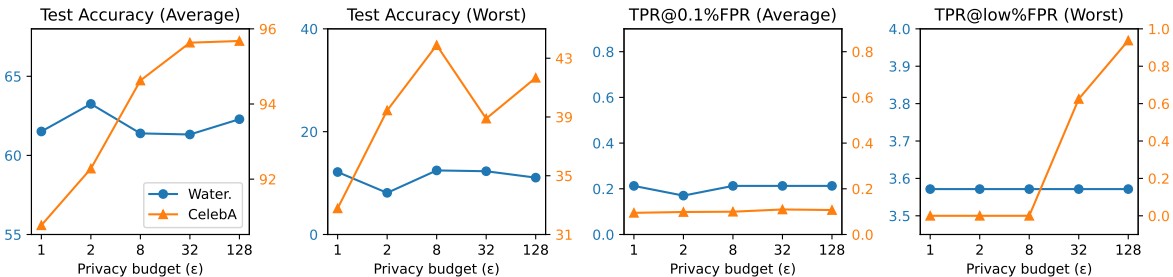

Figure 5: Varying the privacy budget $\epsilon$ in large batch size DP-SGD reveals trade-offs. On CelebA, looser privacy budget $\epsilon$ values yield higher average and worst-group utility, while tighter budgets can improve worst-class privacy protection. This effect is absent on Waterbirds, which is trained from scratch instead of using a pretrained initialization.

literature, a pretrained checkpoint is commonly used, and without it, the model fails at learning generalizable features.

For privacy mitigation, we observe a utility–privacy trade-off in Figure 5 (right). On CelebA, while average privacy remains relatively stable, stricter $\epsilon$ values can improve privacy protection for the worst group. In contrast, Waterbirds shows no significant change, with both the average and worst-class TPR remaining consistent across $\epsilon$. In summary, while differential privacy can improve utility fairness and privacy protection, its effectiveness depends on the presence of a DP suitable pretrained model (Tramèr et al., 2024) to also mitigate spurious correlation.

## 5 Architecture influence on spurious correlations and privacy

Prior privacy works mostly focused on settings with ResNet-like architecture. However, modern architectures have different components that impact feature learning (e.g. masking from He et al. (2022) or attention from Dosovitskiy et al. (2021)). We evaluate models that are sufficiently "diverse", comparing models from different families (convolution and transformers), with different pretraining strategies (supervised and self-supervised), and released at different times (e.g. ResNet and its successor ConvNext). We provide *counterexamples* that revisit two prior research findings. First, vision transformers are not more robust than convolution models under a fair spurious evaluation setup. Second, the best shadow architecture does not always match the target architecture.

Table 2: Target model architecture accuracy on Waterbirds. Modern architectures are better at mitigating spurious correlation compared to older ones. However, the best performing convolutional and transformer based models show no significant difference in worst-group accuracy.

| Model | Train Acc. | Test Acc. | Test WGA |
|---|---|---|---|
| ResNet50 | 96.94 ± 0.03 | 81.08 ± 0.25 | 34.42 ± 0.43 |
| BiT-S | 96.73 ± 0.07 | 79.90 ± 0.21 | 42.37 ± 0.89 |
| CNext-T | 97.47 ± 0.04 | 83.36 ± 0.32 | 47.73 ± 0.97 |
| CNextV2-T | 98.33 ± 0.09 | **83.96 ± 0.22** | 51.90 ± 1.38 |
| ViT-S | 97.46 ± 0.07 | 80.76 ± 0.20 | 43.68 ± 0.60 |
| Deit3-S | 97.27 ± 0.05 | 83.66 ± 0.15 | 51.46 ± 0.49 |
| Swin-T | **98.43 ± 0.07** | 83.72 ± 0.36 | **52.06 ± 1.30** |
| Hiera-T | 98.27 ± 0.09 | 82.60 ± 0.37 | 43.58 ± 0.83 |

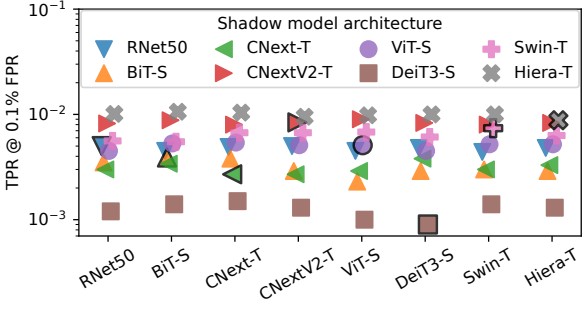

Figure 6: Varying the target and shadow architecture on Waterbirds. With spurious data, the most successful attack is not when the shadow correctly guesses the target.

*Experiment setup.* All the models used are pretrained using the state-of-the-art recipe on the ImageNet1K dataset from the timm library. We compare ResNet50 (He et al., 2016), BiT-S (Kolesnikov et al., 2020), CNext-T (Liu et al., 2022b), CNextV2-T (Woo et al., 2023), ViT (Dosovitskiy et al., 2021), Swin-T (Liu et al., 2021b), DeiT-S (Touvron et al., 2022), and Hiera (Ryali et al., 2023) on Waterbirds. To ensure a fair comparison, all the models have a similar number of parameters, ranging from 20M to 30M. We train 16 shadow models for each architecture as in Section 3 while monitoring the train-validation loss to prevent overfitting. Appendix Table 8 reports the details about the grid search and summarizes the best hyperparameters. The results are averaged over 16 seeds.

**Are ViTs more spurious robust than CNNs?** Ghosal & Li (2024) claim that vision transformers are more robust than convolutional models. We revisit this statement, showing that under the same grid search, both architectures perform comparably. Table 2 shows that the best convolutional and transformer based models (Swin and CNextV2) have similar WGA of $\approx 52\%$. Moreover, our grid search on the architecture DeiT-S/16 achieves $+4.7\%$ absolute improvement in WGA over the value reported by Ghosal & Li (2024) (Table 4), raising concerns about their evaluation protocol. We also note that they unfairly compare the pretrained convolution model BiT with ViT/DeiT, since the latter is pretrained on ImageNet1K with a complex optimization recipe (RandAug, Mixup, CutMix, LabelSmoothing, StochasticDepth, LayerScale) while BiT is not. Our benchmark addresses these issues by comparing a broader set of architectures under a fixed compute budget, ensuring fair evaluation. Additional consistent results are reported for the CelebA dataset in the Appendix Table 9.

> **Finding IV.** *Convolutional models are similarly robust as vision transformers under a fair experimental setup (revisiting Ghosal & Li (2024)).*

**Pretraining recipe and architecture matter for privacy auditing.** Prior works audit the privacy of different architectures on well-balanced datasets such as CIFAR or ImageNet (Shokri et al., 2017; Carlini et al., 2022; Liu et al., 2022a). Our setup compares state-of-the-art architectures on non-balanced datasets Waterbirds, showing that architecture choice can matter for privacy auditing. Using LiRA, we run the attack through all the permutations of shadow/target architecture pairs, resulting in 64 different configurations. Figure 6 shows that when we fix the shadow architecture and vary the target, there is no particular architecture that is more resistant than others. On the contrary, when fixing the target architecture while varying the shadow, we observe that some architectures consistently achieve the strongest attack across all the targets. Moreover, despite sharing the exact same architecture with ViT but differing in optimization recipe, DeiT is consistently ranked as the lowest, suggesting that *the pretraining optimization can greatly influence the attack success of shadow models.* Lastly, prior works reported that the most successful attack is when the shadow matches the target architecture (Carlini et al., 2022; Liu et al., 2022a). In our setup with spurious data, we do not observe the same trend. In Figure 6, the black-outlined markers represent the match between shadow and target architectures, which corresponds to a suboptimal attack. Additional consistent results are reported for the CelebA dataset in the Appendix Figure 10.

> **Finding V.** *The best choice for the shadow architecture does not always match the target when trained on spurious correlated data (extending Carlini et al. (2022) and Liu et al. (2022a)).*

Lastly, further results in the Appendix (Figures 9 and 11) show that all the evaluated architectures exhibit spurious privacy leakage. Across architectures (e.g. Resnet, Convnext, ViT, and Swin), the difference between the most and the least spurious group is approximately one order of magnitude under TPR at 0.1% FPR (see Figure 9 groups 2/3 for Waterbirds and Figure 11 groups 3/1 for CelebA). However, note that the privacy leakage across all the different target architectures is at a similar level, suggesting that architecture choice alone plays a small role in mitigating spurious privacy leakage.

# 6 Conclusion

Our findings highlight critical privacy concerns when training neural networks on real-world datasets with spurious correlations. The existence of *spurious privacy leakage* makes spurious data significantly more vulnerable to privacy attacks than non-spurious data. Using this information, an adversary can craft better-tailored attacks against specific demographic groups. To detect and prevent this, the auditor can go beyond aggregate metrics and include a fine-grained group-level analysis of both performance and privacy fairness. This is particularly important for datasets with simpler task objectives (Section 3.2).

We further show that existing strategies are not sufficient. For example, the most promising approaches of *spurious robust methods fail to address spurious privacy leakage*. Our analysis attributes the failure to high feature similarity between robust and standard training models, demonstrating that the underlying memorization behavior remains unchanged. Moreover, alternative spurious methods (Section 2) effectively reduce to variants of DRO or DFR relying on group resampling or sample reweighting and therefore are likely to show similar limitations. Although differential privacy offers a formal guarantee, a tight privacy budget comes at the expense of the worst-group utility, which is not practical.

Our results open promising directions for future research. On the attack side, privacy disparity enables adversaries to achieve higher MIA success by targeting only subgroups of data. For example, an adversary can inject poisoned samples (Tramèr et al., 2022) to strengthen a spurious correlation, allowing the attacker to target a subgroup with a higher success rate. Another possible direction is to improve the auditing of spurious data in real world without explicit spurious attribute labels. For example, label-free spurious robust methods (Yang et al., 2024) can be used to detect spurious correlations and perform auditing of both performance and privacy fairness, mitigating undesired deployment.

**Limitations.** Our results are based on a state-of-the-art attack-based evaluation rather than analytical guarantees. Although our approach is more practical and provides empirical evidence, it is limited to the choice of our experiment settings, which include five real-world datasets, eight model architectures, and two robust training methods.

**Reproducibility.** We have made our code publicly available through an anonymized repository (Section 1). Details for each experiment setup are presented in the respective sections, along with the corresponding grid search and the best hyperparameters in the Appendix (see Tables 3 and 8). The spurious data sets are presented in Appendix A. The code is released at https://github.com/orientino/spurious-mia.

## Acknowledgments

CZ is funded by the research project R-STR-8019-00-B from the University of Luxembourg. The experiments were supported by the ULHPC facilities from the University of Luxembourg. [2]

---

[2] https://hpc-docs.uni.lu/

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

## Appendix

We report the dataset details, additional results on group privacy disparity, a comparison of different membership inference attack methods, define and show the memorization score for each dataset, and more results on differential privacy and model architectures.

## A  Dataset

**Waterbirds** Sagawa et al. (2019). Vision dataset where the task is to classify whether landbird or waterbird. The background is the spurious feature represented as water or land background. The presence of the spurious features induces four data groups: landbird on land background, landbird on water background, waterbird on water background, and waterbird on land background. The groups have respectively 3498, 184, 1057, and 56 samples. Therefore, the type of bird is spurious correlated with the same type of background.

**CelebA** Liu et al. (2015). Vision dataset where the task is to classify whether a celebrity is a male or female. The hair color is the spurious features represented as dark or blonde hair. The presence of spurious features induces four data groups: female with blonde hair, female with dark hair, male with dark hair, and male with blonde hair. The groups have respectively 71629, 66874, 22880, and 1387 samples. Therefore, blonde hair is spurious correlated with female celebrities.

**FMoW** Koh et al. (2021). Vision dataset where the task is to identify between 62 classes the type of land usage, e.g. hospital, airport, single or multi-use residential area. The geographical location is the spurious feature representing the continents: Asia, Europe, Africa, Americas, and Oceania. The groups have respectively 17809, 34816, 1582, 20973, and 1641 samples whereas the African countries have the majority of samples as single-use residential areas (36%). Therefore, samples collected from Africa are spurious correlated with the single-unit residential areas. Moreover, the test set presents a distribution shift with samples collected from different years.

**MultiNLI** Williams et al. (2017). Text dataset where the task is to identify the relationship between two pairs of text as a contradiction, entailment, or neither. The negation is the spurious feature usually found in the contradiction class. The presence of the spurious feature induces six data groups: contradiction without negation, contradiction with negation, entailment without negation, entailment with negation, neutral without negation, and neutral with negation. The groups have respectively 57498, 11158, 67376, 1521, 66630, and 1991 samples. Therefore, samples with the spurious feature negation are correlated with the contradiction class.

**CivilComments** Koh et al. (2021). Text dataset with the task of detecting toxic comments of online articles. The demographic identities (male, female, LGBTQ, Christian, Muslim, other religions, Black, and White) combined with the target (toxic or not) divides the dataset into 16 groups groups. The group "other religions" is spurious correlated with the target. The groups have respectively 16568, 26846, 5638, 27824, 11064, 4402, 4727, 9812, 2435, 3928, 1865, 1867, 2964, 608, 2076, and 3462 samples.

## B  Spurious privacy leakage

We report additional technical details related to Section 3 and include additional results: formalizing the feature similarity and complexity definitions, comparing different membership inference attacks on spurious data, demonstrating how memorization of spurious data causes higher privacy leakage.

*Hyperparameters.* For Section 3, we apply grid search to find the best hyperparameters for each dataset. For Waterbirds and CelebA we search the learning rate between [1e-3, 1e-4] and weight decay [1e-1, 1e-2, 1e-3]. For FMoW the learning rate [1e-3, 3e-3, 1e-4, 3e-4], weight decay [1e-1, 1e-2, 1e-3], and epochs [20, 30, 40]. For MultiNLI the learning rate [1e-5, 3e-5], weight decay [1e-5, 1e-4]. For CivilComments the learning rate [1e-5, 1e-6], weight decay [1e-3, 1e-4]. The best hyperparameters are reported at Table 3.

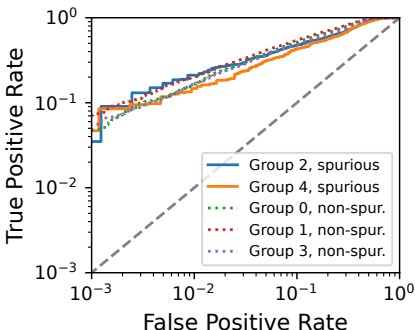

Figure 7: Spurious privacy leakage is absent for the FMoW dataset. See the Section 3.2.

Table 3: Hyperparameters used to train shadow models for each dataset. Adapted from the hyperparameters of Izmailov et al. (2022). Since we trained the models using LiRA algorithm with 50% of the total dataset, we had to grid search and validate on the validation set.

| Data | Optim | Batch size | LR | WD | Epochs | C |
|------|-------|-----------|-----|-----|--------|---|
| Waterbirds | SGD | 32 | 1e-3 | 1e-2 | 100 | 1 |
| CelebA | SGD | 32 | 1e-3 | 1e-2 | 20 | 5 |
| FMoW | SGD | 32 | 3e-3 | 1e-2 | 20 | 1 |
| MultiNLI | AdamW | 16 | 1e-5 | 1e-4 | 5 | 8 |
| CivilComments | AdamW | 32 | 1e-5 | 1e-4 | 5 | 8 |

## B.1 Measuring task complexity

We formalize the measures used in Section 3.2 to quantity the feature similarity and feature complexity between different groups of data.

**Definition B.1** (Feature Similarity). Let $\mathbf{E}_1 \in \mathbb{R}^{n_1 \times d}$ and $\mathbf{E}_2 \in \mathbb{R}^{n_2 \times d}$ denote two feature embeddings, where $n_1$ and $n_2$ are the number of samples and $d$ is the feature dimension. The feature similarity is quantified using the linear centered kernel alignment (CKA) score (Kornblith et al., 2019), which is invariant to orthogonal transformations and isotropic scaling:

$$\mathrm{CKA}(\mathbf{E}_1, \mathbf{E}_2) = \frac{\mathrm{HSIC}(\mathbf{E}_1, \mathbf{E}_2)}{\sqrt{\mathrm{HSIC}(\mathbf{E}_1, \mathbf{E}_1) \cdot \mathrm{HSIC}(\mathbf{E}_2, \mathbf{E}_2)}},$$

where $\mathrm{HSIC}(\mathbf{A}, \mathbf{B}) = \mathrm{tr}(\mathbf{K_A} \mathbf{H} \mathbf{K_B} \mathbf{H})$ is the Hilbert-Schmidt Independence Criterion, $\mathbf{K_A}$ and $\mathbf{K_B}$ are the Gram matrices of $\mathbf{A}$ and $\mathbf{B}$, and $\mathbf{H}$ is the centering matrix.

**Definition B.2** (Feature Complexity). Let $\mathbf{E} \in \mathbb{R}^{n \times d}$ represent the feature embeddings with $n$ samples and $d$ dimensions. The feature complexity is the smallest number of principal components $k$ required to explain $\tau$ cumulative variance in $\mathbf{E}$, defined as:

$$k = \min\left\{k' : \mathrm{EVR}(k') \geq \tau\right\}, \quad \mathrm{EVR}(k) = \frac{\sum_{i=1}^{k} \lambda_i}{\sum_{j=1}^{d} \lambda_j}.$$

where the explained variance ratio (EVR) is the cumulative variance of the first $k$ principal components over the total variance, and $\lambda_i$ are the eigenvalues of the covariance matrix of $\mathbf{E}$.

## B.2 Membership inference attacks comparison

Most of the previous MIAs are limited by the assumption that all the samples have the same level of importance (or hardness) (Yeom et al., 2018; Shokri et al., 2017), which is incorrect since natural data follow

a long-tail distribution (Feldman, 2020). We compare three different state-of-the-art MIAs and show that the phenomenon of *spurious privacy leakage* exists regardless of the attack used. We use two different versions of LiRA (Carlini et al., 2022), online and offline, and TrajMIA (Liu et al., 2022a). The results in Table 4 show that all the methods successfully reveal the disparity on Waterbirds, and LiRA online is the strongest attack on vulnerable groups.

Table 4: Comparing the attack success rate of different membership inference attacks on ERM models trained with Waterbirds. All the methods can be used to identify the privacy disparity, but LiRA poses a greater risk for more vulnerable spurious groups. *TPRs are reported at ~1% and ~3% for groups 1 and 2 respectively due to their limited sample size. The spurious groups are highlighted .

| | TPR @ 0.1% FPR (↑) | | | AUROC (↑) | | |
| Group | LiRA | LiRA (offline) | TrajMIA | LiRA | LiRA (offline) | TrajMIA |
|---|---|---|---|---|---|---|
| 1 | $0.22 \pm 0.03$ | $0.14 \pm 0.02$ | $\mathbf{1.67 \pm 3.27}$ | $51.78 \pm 0.15$ | $49.97 \pm 0.22$ | $\mathbf{58.20 \pm 3.42}$ |
| 2* | $\mathbf{10.87 \pm 1.18}$ | $5.39 \pm 0.78$ | $3.18 \pm 0.47$ | $\mathbf{75.07 \pm 0.54}$ | $61.32 \pm 1.01$ | $70.28 \pm 1.22$ |
| 3* | $\mathbf{30.91 \pm 2.81}$ | $18.98 \pm 2.13$ | $14.60 \pm 1.69$ | $85.83 \pm 0.76$ | $69.50 \pm 1.67$ | $\mathbf{86.16 \pm 2.55}$ |
| 4 | $1.73 \pm 0.19$ | $0.83 \pm 0.11$ | $\mathbf{6.57 \pm 0.59}$ | $60.52 \pm 0.34$ | $53.63 \pm 0.42$ | $\mathbf{72.40 \pm 2.31}$ |
| T | $1.16 \pm 0.07$ | $0.44 \pm 0.04$ | $\mathbf{1.68 \pm 0.00}$ | $55.44 \pm 0.14$ | $51.43 \pm 0.16$ | $\mathbf{74.74 \pm 0.00}$ |

### B.3 Memorization score of spurious groups

Feldman (2020) introduced the notion of label memorization (Definition B.3) as the difference in the label of a model trained with or without $\boldsymbol{x}$. We use the models from the LiRA algorithm from Section 3.1 to approximate the memorization score. Carlini et al. (2022) proposed the privacy score $d = |\mu_{\text{in}} - \mu_{\text{out}}|/(\sigma_{\text{in}} + \sigma_{\text{out}})$ to measure the difference between the loss distributions coming from IN and OUT shadow models of LiRA. Note that both mem(.) and $d$ measure the difference between two probability distributions conditioned on $D$ and $D \setminus \{i\}$ but with a different level of granularity; label memorization is coarser than $d$ and collapses the whole distributions to a single scalar, the probability of outputting the correct label.

**Definition B.3** (Label memorization). Label memorization is the difference in the output label of a model $f \sim \mathcal{A}(\mathcal{D})$ fit on the dataset $D$ with or without a specific data point $(\boldsymbol{x}_i, \boldsymbol{y}_i) \sim D$. Formally, $\text{mem}(\mathcal{A}, D, i) = \left| \Pr_{f \sim \mathcal{A}(D)} (f(\boldsymbol{x}_i) = \boldsymbol{y}_i) - \Pr_{f \sim \mathcal{A}(D \setminus \{i\})} (f(\boldsymbol{x}_i) = \boldsymbol{y}_i) \right|$

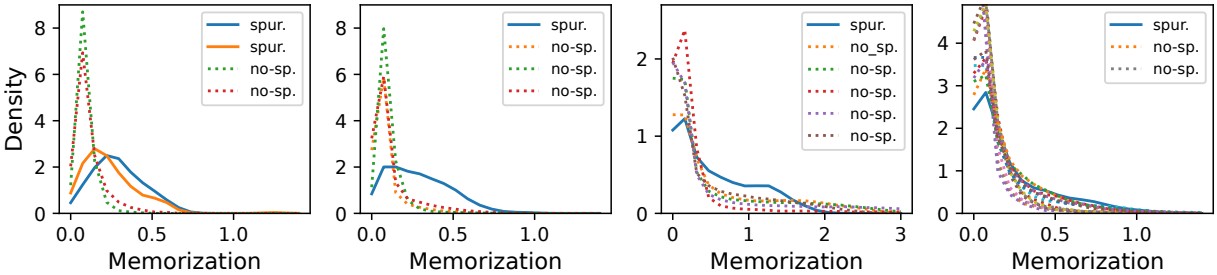

Figure 8: Memorization score divided per group on Waterbirds, CelebA, MultiNLI, and CivilComments respectively. Spurious correlated groups (solid lines) have on average a higher memorization score than non-spurious groups, which indicates that models treat spurious groups as atypical examples. In the FMoW dataset, all the groups have similar levels of memorization.

We compute $d$ for each data point and use a Gaussian kernel density estimator to fit each group. The results in Figure 8 show the estimated frequency of the memorization score for the whole dataset divided per group. We observe that the spurious groups have, on average, higher memorization scores compared to non-spurious groups (except for FMoW as in Figure 1). The increase can be attributed to the presence of spurious features, which turn typical examples into atypical ones that the model has to memorize. A higher memorization score is known to be linked to a higher vulnerability under privacy attacks (Feldman, 2020), which matches what we observed previously.

Table 5: Evaluating the different training methods across datasets. DRO and DFR consistently mitigate spurious features by reducing the gap between train-test WGA compared to ERM. After an extensive grid search, DRO fails to improve the validation WGA on FMoW, therefore we omit it.

| Data | Model | Train Acc. (↑) | Test Acc. (↑) | △Acc. (↓) | Train WGA (↑) | Test WGA (↑) | △WGA (↓) |
|---|---|---|---|---|---|---|---|
| Water. | ERM | **97.16 ± 0.11** | 81.12 ± 0.35 | 16.0 | 50.18 ± 2.70 | 34.30 ± 1.27 | 15.8 |
| | DRO | 96.16 ± 0.23 | **86.42 ± 0.38** | 9.7 | **93.73 ± 0.44** | 78.12 ± 0.84 | 15.6 |
| | DFR | 92.63 ± 1.13 | 85.98 ± 0.60 | **6.7** | 85.81 ± 1.95 | **77.67 ± 2.13** | **8.2** |
| CelebA | ERM | **97.12 ± 0.03** | **95.82 ± 0.06** | 1.3 | 62.81 ± 1.82 | 42.67 ± 0.62 | 20.2 |
| | DRO | 94.47 ± 0.05 | 93.23 ± 0.21 | **1.2** | **91.84 ± 0.36** | **86.11 ± 0.89** | 5.7 |
| | DFR | 95.43 ± 0.14 | 90.52 ± 0.22 | 4.9 | 89.46 ± 0.36 | 84.00 ± 0.60 | **5.4** |
| Multi. | ERM | **97.26 ± 0.04** | 80.74 ± 0.04 | 16.5 | **91.43 ± 0.79** | 61.76 ± 0.28 | 29.7 |
| | DRO | 89.69 ± 0.09 | 78.76 ± 0.07 | **10.9** | 85.34 ± 0.23 | **72.96 ± 0.66** | **12.4** |
| | DFR | 96.36 ± 0.14 | **79.17 ± 0.06** | 17.2 | 90.84 ± 0.11 | 71.33 ± 0.13 | 19.5 |
| Civil. | ERM | **97.45 ± 0.04** | **88.03 ± 0.06** | 9.4 | **90.41 ± 0.32** | 53.26 ± 0.59 | 37.1 |
| | DRO | 90.00 ± 0.29 | 81.11 ± 0.31 | 8.9 | 81.84 ± 0.85 | 68.60 ± 0.47 | 13.2 |
| | DFR | 86.88 ± 0.17 | 79.38 ± 0.05 | **7.5** | 77.49 ± 0.69 | **69.47 ± 0.26** | **8.0** |
| FMoW | ERM | 91.58 ± 0.04 | **50.85 ± 0.08** | **40.7** | **90.84 ± 0.06** | 31.04 ± 0.20 | 59.8 |
| | DRO | - | - | - | - | - | - |
| | DFR | **91.20 ± 0.38** | 48.62 ± 0.09 | 42.6 | 88.57 ± 0.55 | **32.44 ± 0.34** | **56.1** |

# C   Robust training

We report additional technical details related to Section 4.1 and include an additional result analyzing the privacy side effect of choosing L2 vs L1 regularization in DFR.

*Hyperparameters.* We use the same hyperparameters as in Table 3. Robust training DRO requires an extra hyperparameter C. For Waterbirds and CelebA we tune C within [0, 1, 2, 3, 4], for FMoW [0, 1, 2, 4, 8, 16], and for MultiNLI and CivilComments [0, 1, 2, 4, 8, 16]. For DFR, we do not use the validation set for retraining but use a group-balanced subset sampled from the training set. This allows a fairer comparison with other methods by not exploiting additional data, and it is also necessary for a fair privacy analysis since adding extra data invalidates the membership inference comparison.

*Experiment setup.* For the LiRA attack, we train 32 ERM shadow models for Waterbirds and CelebA and 16 ERM shadow models for FMoW, MultiNLI, and CivilComments. We also train 5 DRO and DFR models for all the datasets. We use the online version of LiRA with a fixed variance for all the attacks to audit the privacy level. Table 1 reports the mean and standard error of using the ERM trained shadow models to attack 32 target models for each training type of Waterbirds and CelebA, and 5 for FMoW and MultiNLI.

We found DRO to be unstable on more complex datasets such as FMoW, where it fails to improve the validation WGA even after an extensive hyperparameter grid search. While for DFR, despite its simplicity and effectiveness compared to DRO, we find that it can slightly increase the vulnerability of spurious groups. However, by simply changing DFR's regularization from L1 to L2 norm, we achieve an accuracy-privacy tradeoff reducing the vulnerability to the same level as ERM at the cost of a lower WGA (77.67% to 73.00%) (see Appendix C.1).

## C.1   DFR with L2 regularization

DFR with the L1 regularization achieves the best performance measured with WGA. The L1 regularization encourages sparsity of the last-linear layer, concentrating most of the weights to 0. Kirichenko et al. (2022) showed that using L2 regularization leads to suboptimal results in terms of WGA performance. Additionally, we find that L2 leads to an accuracy-privacy tradeoff where it slightly increases privacy protection against MIA. We compare DFR trained with L2 and the default L1 regularization, finding that L2 regularization achieves a lower 83.65% test accuracy compared to 85.96% of L1, and also a lower WGA 73.00% compared to 77.67%. However, in Table 7, we observe that by using L2 regularization, the privacy vulnerability is reduced to a similar level of ERM.

Table 6: Comparing the attack success rate of different training methods for spurious and non-spurious groups. This it the full version of the Table 1 in the main text. The spurious groups are  highlighted . *The spurious groups 1 and 2 of Waterbirds are evaluated at 1% and 3% respectively due to the limited samples. DRO fails to improve the accuracy on FMoW after an extensive grid search, therefore we omit it.

| Data | Group (n) | TPR @ 0.1% FPR (↓) | | | AUROC (↓) | | |
|---|---|---|---|---|---|---|---|
| | | ERM | DRO | DFR | ERM | DRO | DFR |
| Waterb. | 0 (1749) | $0.22 \pm 0.03$ | $0.22 \pm 0.03$ | $0.22 \pm 0.03$ | $51.78 \pm 0.15$ | $\mathbf{51.59 \pm 0.16}$ | $51.64 \pm 0.16$ |
| | 1 (92)* | $\mathbf{10.87 \pm 1.18}$ | $10.91 \pm 1.08$ | $11.16 \pm 1.20$ | $75.07 \pm 0.54$ | $\mathbf{74.69 \pm 0.58}$ | $75.15 \pm 0.52$ |
| | 2 (28)* | $\mathbf{30.91 \pm 2.81}$ | $31.06 \pm 2.76$ | $33.20 \pm 2.83$ | $85.83 \pm 0.76$ | $\mathbf{85.54 \pm 0.77}$ | $86.17 \pm 0.79$ |
| | 3 (528) | $\mathbf{1.73 \pm 0.19}$ | $1.73 \pm 0.19$ | $1.91 \pm 0.20$ | $60.52 \pm 0.34$ | $\mathbf{60.33 \pm 0.42}$ | $60.66 \pm 0.33$ |
| | T (2397) | $1.16 \pm 0.07$ | $\mathbf{1.13 \pm 0.06}$ | $1.19 \pm 0.06$ | $55.44 \pm 0.14$ | $\mathbf{55.23 \pm 0.17}$ | $55.39 \pm 0.15$ |
| CelebA | 0 (35814) | $0.53 \pm 0.01$ | $\mathbf{0.51 \pm 0.02}$ | $0.52 \pm 0.02$ | $53.12 \pm 0.05$ | $\mathbf{52.89 \pm 0.15}$ | $53.04 \pm 0.11$ |
| | 1 (33437) | $0.27 \pm 0.01$ | $\mathbf{0.26 \pm 0.01}$ | $\mathbf{0.26 \pm 0.01}$ | $50.58 \pm 0.05$ | $\mathbf{50.48 \pm 0.10}$ | $50.56 \pm 0.06$ |
| | 2 (11440) | $1.64 \pm 0.05$ | $\mathbf{1.58 \pm 0.06}$ | $1.62 \pm 0.05$ | $59.77 \pm 0.08$ | $59.44 \pm 0.26$ | $\mathbf{59.36 \pm 0.26}$ |
| | 3 (693) | $4.61 \pm 0.50$ | $\mathbf{4.56 \pm 0.48}$ | $4.77 \pm 0.46$ | $80.51 \pm 0.21$ | $\mathbf{79.95 \pm 0.52}$ | $80.00 \pm 0.48$ |
| | T (81384) | $0.76 \pm 0.01$ | $\mathbf{0.73 \pm 0.02}$ | $0.74 \pm 0.01$ | $53.43 \pm 0.04$ | $\mathbf{53.22 \pm 0.14}$ | $53.30 \pm 0.11$ |
| MultiNLI | 0 (14374) | $6.95 \pm 0.66$ | $\mathbf{6.65 \pm 0.61}$ | $6.78 \pm 0.62$ | $74.36 \pm 0.36$ | $\mathbf{74.23 \pm 0.40}$ | $74.26 \pm 0.34$ |
| | 1 (2789) | $\mathbf{2.03 \pm 0.21}$ | $2.12 \pm 0.22$ | $2.13 \pm 0.21$ | $\mathbf{56.81 \pm 1.21}$ | $56.95 \pm 1.37$ | $56.98 \pm 1.36$ |
| | 2 (16844) | $5.88 \pm 0.42$ | $\mathbf{5.73 \pm 0.45}$ | $5.86 \pm 0.40$ | $72.04 \pm 0.31$ | $\mathbf{71.93 \pm 0.30}$ | $72.01 \pm 0.30$ |
| | 3 (380) | $6.14 \pm 1.84$ | $\mathbf{5.66 \pm 1.73}$ | $6.22 \pm 1.84$ | $77.41 \pm 0.33$ | $77.28 \pm 0.27$ | $\mathbf{77.25 \pm 0.30}$ |
| | 4 (16657) | $5.81 \pm 0.25$ | $\mathbf{5.67 \pm 0.25}$ | $5.85 \pm 0.28$ | $75.83 \pm 0.15$ | $\mathbf{75.64 \pm 0.20}$ | $75.67 \pm 0.13$ |
| | 5 (498) | $8.26 \pm 0.55$ | $9.08 \pm 1.37$ | $\mathbf{7.78 \pm 0.57}$ | $83.70 \pm 0.53$ | $\mathbf{83.49 \pm 0.61}$ | $83.59 \pm 0.57$ |
| | T (51542) | $5.95 \pm 0.42$ | $\mathbf{5.83 \pm 0.44}$ | $5.93 \pm 0.42$ | $73.44 \pm 0.16$ | $\mathbf{73.31 \pm 0.19}$ | $73.33 \pm 0.13$ |
| CivilCom. | 0 (8284) | $0.13 \pm 0.02$ | $0.12 \pm 0.02$ | $\mathbf{0.10 \pm 0.02}$ | $\mathbf{50.46 \pm 0.14}$ | $50.59 \pm 0.36$ | $50.55 \pm 0.36$ |
| | 1 (13423) | $0.12 \pm 0.03$ | $\mathbf{0.12 \pm 0.02}$ | $0.13 \pm 0.02$ | $\mathbf{50.38 \pm 0.22}$ | $50.68 \pm 0.47$ | $50.68 \pm 0.46$ |
| | 2 (2819) | $0.16 \pm 0.04$ | $\mathbf{0.11 \pm 0.03}$ | $0.13 \pm 0.03$ | $\mathbf{49.79 \pm 0.38}$ | $50.19 \pm 0.26$ | $49.97 \pm 0.24$ |
| | 3 (13912) | $\mathbf{0.10 \pm 0.02}$ | $0.14 \pm 0.03$ | $0.12 \pm 0.03$ | $50.60 \pm 0.25$ | $50.44 \pm 0.24$ | $\mathbf{50.40 \pm 0.24}$ |
| | 4 (5532) | $\mathbf{0.11 \pm 0.01}$ | $0.12 \pm 0.02$ | $0.13 \pm 0.01$ | $\mathbf{50.09 \pm 0.18}$ | $50.55 \pm 0.22$ | $50.56 \pm 0.22$ |
| | 5 (2201) | $0.23 \pm 0.07$ | $0.30 \pm 0.09$ | $\mathbf{0.18 \pm 0.05}$ | $\mathbf{50.59 \pm 0.44}$ | $50.82 \pm 0.42$ | $50.84 \pm 0.38$ |
| | 6 (2363) | $0.08 \pm 0.06$ | $\mathbf{0.07 \pm 0.05}$ | $0.08 \pm 0.06$ | $\mathbf{49.60 \pm 0.25}$ | $50.03 \pm 0.22$ | $50.18 \pm 0.21$ |
| | 7 (4906) | $0.11 \pm 0.02$ | $0.13 \pm 0.04$ | $\mathbf{0.09 \pm 0.02}$ | $50.07 \pm 0.36$ | $\mathbf{50.00 \pm 0.28}$ | $50.08 \pm 0.22$ |
| | 8 (1217) | $0.20 \pm 0.07$ | $\mathbf{0.11 \pm 0.05}$ | $0.14 \pm 0.07$ | $50.03 \pm 0.34$ | $\mathbf{49.55 \pm 0.59}$ | $50.22 \pm 0.49$ |
| | 9 (1964) | $0.21 \pm 0.06$ | $\mathbf{0.16 \pm 0.04}$ | $0.18 \pm 0.04$ | $49.99 \pm 0.44$ | $49.91 \pm 0.22$ | $\mathbf{49.81 \pm 0.39}$ |
| | 10 (932) | $\mathbf{0.11 \pm 0.04}$ | $0.11 \pm 0.04$ | $0.11 \pm 0.04$ | $\mathbf{50.13 \pm 0.86}$ | $50.51 \pm 1.01$ | $50.64 \pm 0.96$ |
| | 11 (933) | $\mathbf{0.07 \pm 0.04}$ | $0.21 \pm 0.12$ | $0.24 \pm 0.11$ | $49.31 \pm 0.69$ | $48.87 \pm 0.61$ | $\mathbf{48.75 \pm 0.60}$ |
| | 12 (1482) | $0.02 \pm 0.02$ | $\mathbf{0.00 \pm 0.00}$ | $0.00 \pm 0.00$ | $49.66 \pm 0.24$ | $50.23 \pm 0.44$ | $\mathbf{49.84 \pm 0.33}$ |
| | 13 (304) | $0.44 \pm 0.10$ | $0.43 \pm 0.17$ | $\mathbf{0.32 \pm 0.12}$ | $50.72 \pm 1.70$ | $50.91 \pm 1.86$ | $\mathbf{50.53 \pm 1.71}$ |
| | 14 (1038) | $0.22 \pm 0.11$ | $\mathbf{0.19 \pm 0.11}$ | $0.26 \pm 0.12$ | $51.84 \pm 0.59$ | $51.84 \pm 0.63$ | $\mathbf{51.80 \pm 0.56}$ |
| | 15 (1731) | $0.38 \pm 0.10$ | $0.26 \pm 0.07$ | $\mathbf{0.20 \pm 0.09}$ | $49.78 \pm 0.48$ | $50.25 \pm 0.62$ | $\mathbf{49.76 \pm 0.47}$ |
| | T (38018) | $\mathbf{0.09 \pm 0.01}$ | $0.09 \pm 0.01$ | $0.09 \pm 0.01$ | $\mathbf{50.24 \pm 0.07}$ | $50.41 \pm 0.11$ | $50.38 \pm 0.09$ |
| FMoW | 0 (8904) | $\mathbf{5.14 \pm 0.41}$ | - | $5.22 \pm 0.37$ | $83.70 \pm 0.05$ | - | $\mathbf{83.60 \pm 0.05}$ |
| | 1 (17408) | $\mathbf{7.45 \pm 0.27}$ | - | $7.61 \pm 0.28$ | $85.12 \pm 0.07$ | - | $\mathbf{84.96 \pm 0.08}$ |
| | 2 (791) | $\mathbf{6.30 \pm 1.80}$ | - | $6.42 \pm 1.80$ | $81.54 \pm 0.22$ | - | $81.62 \pm 0.24$ |
| | 3 (10486) | $\mathbf{5.53 \pm 0.31}$ | - | $5.69 \pm 0.32$ | $82.85 \pm 0.13$ | - | $\mathbf{82.74 \pm 0.13}$ |
| | 4 (820) | $\mathbf{4.61 \pm 0.95}$ | - | $5.34 \pm 0.88$ | $80.37 \pm 0.45$ | - | $\mathbf{80.26 \pm 0.52}$ |
| | T (38409) | $6.54 \pm 0.21$ | - | $\mathbf{6.47 \pm 0.17}$ | $84.02 \pm 0.05$ | - | $83.90 \pm 0.03$ |

Table 7: Comparing DFR L2 and L1 regularization under the LiRA attack with 32 ERM shadow models trained on Waterbirds. The results are averaged over 32 target models.

| Group (n) | TPR @ low% FPR | | |
|---|---|---|---|
| | ERM | DFR L1 | DFR L2 |
| 0 (1749) | $0.22 \pm 0.03$ | $0.22 \pm 0.03$ | $0.22 \pm 0.03$ |
| 1 (92) | $10.87 \pm 1.18$ | $11.16 \pm 1.20$ | $\mathbf{10.84 \pm 1.16}$ |
| 2 (28) | $30.91 \pm 2.81$ | $33.20 \pm 2.83$ | $\mathbf{30.52 \pm 2.70}$ |
| 3 (528) | $1.73 \pm 0.19$ | $1.91 \pm 0.20$ | $\mathbf{1.67 \pm 0.21}$ |
| T (2397) | $1.16 \pm 0.07$ | $1.19 \pm 0.06$ | $\mathbf{1.10 \pm 0.07}$ |

# D  Differential privacy

**Definition D.1** (Differential privacy)**.** A randomized mechanism $\mathcal{M} \colon \mathcal{D} \to \mathcal{R}$ satisfies $(\epsilon, \delta)$-differential privacy if for any two datasets differing by a single data point $D, D' \in \mathcal{D}$ and for any subset of outputs $S \subseteq \mathcal{R}$ it holds that

$$\Pr[\mathcal{M}(D) \in S] \leq e^{\epsilon} \Pr[\mathcal{M}(D') \in S] + \delta.$$

where $\epsilon \geq 0$ and $\delta \geq 0$ are privacy parameters. A higher privacy budget $(\epsilon, \delta)$ results in a better utility but lower protection, while a lower privacy budget guarantees the opposite. The privacy budget $(\epsilon, \delta)$ in DP-SGD Abadi et al. (2016) is controlled by the hyperparameters noise level $\sigma$ added to the gradient and clipping threshold $C$ to clip the maximum norm of the gradient. A higher level of noise leads to a lower privacy budget, and the clipping influences the amount of possible noise to add.

We report additional experimental details for the Section 4.2.

*Experiment setup.* Instead of ResNet, we use ConvNext as target model because it uses layer normalization instead of batch normalization, which is not compatible with DP-SGD. Specifically, batch normalization aggregates the statistics of the batch creating a dependency between samples in a batch, which is a privacy violation. The target models are trained using the grid search with lr in [1, 1e-1, 1e-2, 1e-3], $\epsilon$ in [1, 2, 8, 32, 128], and $\delta = $ 1e-4 for Waterbirds and $\delta = $ 1e-5 for CelebA. Each model is trained up to 100 epochs for Waterbirds and 50 for CelebA, and select best validation WGA checkpoint. To audit the privacy level, 32 ERM ConvNext shadow models are used for Waterbirds and CelebA. We use opacus (Yousefpour et al., 2021) with batch size 1024.

# E    Architecture influences

We report additional technical details related to Section 5.

*Experimet setup.* For each model, we use the same computational budget by performing the grid search with the learning rate [1e-1, 1e-2, 1e-3, 1e-4] and the weight decay in [1e-1, 1e-2, 1e-3] and choose the best performing based on the validation set. For privacy analysis, we avoid overfitting by tuning the number of training epochs to stop the training before reaching 100% training accuracy (Carlini et al., 2022). The best hyperparameters are reported at Table 8.

Table 8: Final hyperparameters used for training the various architecture in Section 5.

| Model | Params (M) | Waterbirds | | | CelebA | | |
| | | LR | WD | Epochs | LR | WD | Epochs |
|---|---|---|---|---|---|---|---|
| ResNet50 | 23.5 | 0.001 | 0.01 | 100 | 0.001 | 0.01 | 20 |
| BiT-S | 23.6 | 0.0001 | 0.01 | 10 | - | - | - |
| CNext-T | 27.8 | 0.001 | 0.01 | 10 | - | - | - |
| CNextV2-T | 27.8 | 0.001 | 0.01 | 5 | 0.001 | 0.001 | 20 |
| ViT-S | 21.6 | 0.0001 | 0.01 | 10 | 0.0001 | 0.01 | 10 |
| Deit3-S | 21.6 | 0.0001 | 0.01 | 10 | - | - | - |
| Swin-T | 27.5 | 0.001 | 0.01 | 10 | 0.0001 | 0.01 | 10 |
| Hiera-T | 27.1 | 0.0001 | 0.01 | 20 | - | - | - |

# F    Compute resources

All the experiments are run on our internal cluster with the GPU Tesla V100 16GB/32GB of memory. We give an estimate of the amount of compute required for each experiment. For Section 3, we trained 96 shadow models for Waterbirds and CelebA, and 48 for FMoW, MultiNLI, and CivilComments which took ~600 hours of computing. We trained 16 shadow models for FMoW4 and FMoW16 which took another ~100 hours. For Section 5, we trained in total 128 shadow models on Waterbirds and CelebA averaging around ~100 hours. Lastly for Section 4.2, we trained 32 ConvNext-t shadow models and 5 target models for about ~100 hours. The full research required additional computing for hyperparameter grid searches, in particular for differential privacy training which is known to be difficult.

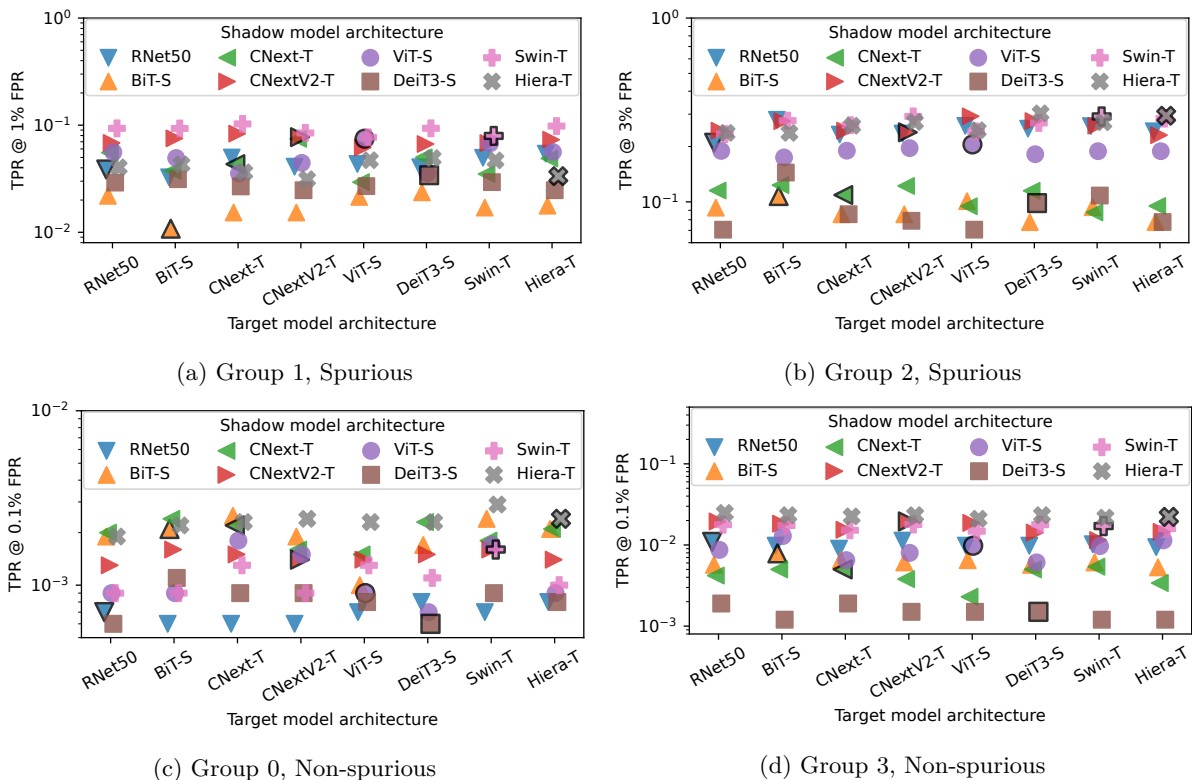

(a) Group 1, Spurious

(b) Group 2, Spurious

(c) Group 0, Non-spurious

(d) Group 3, Non-spurious

Figure 9: Varying the target and shadow model architecture per group on Waterbirds.

Table 9: Target model architecture accuracy on CelebA dataset. Modern architectures are better at mitigating spurious correlation than older ones, with no significant worst-group accuracy difference between the best convolutional and transformer-based architectures.

| Model | Train Acc. | Test Acc. | Test WGA |
|---|---|---|---|
| ResNet50 | $97.12 \pm 0.03$ | $95.82 \pm 0.06$ | $42.67 \pm 0.62$ |
| CNextV2-T | $96.87 \pm 0.08$ | $\mathbf{96.02 \pm 0.03}$ | $\mathbf{44.80 \pm 0.98}$ |
| ViT-S | $\mathbf{98.45 \pm 0.02}$ | $95.75 \pm 0.03$ | $42.99 \pm 0.36$ |
| Swin-T | $98.27 \pm 0.15$ | $95.79 \pm 0.02$ | $44.60 \pm 0.95$ |

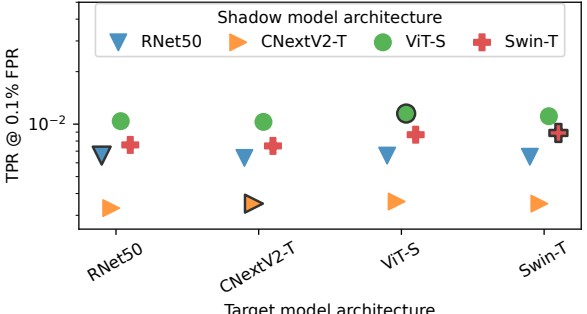

Figure 10: Varying the target and shadow architecture on the whole CelebA dataset. The most successful attack is not when the shadow correctly guesses the target architecture.

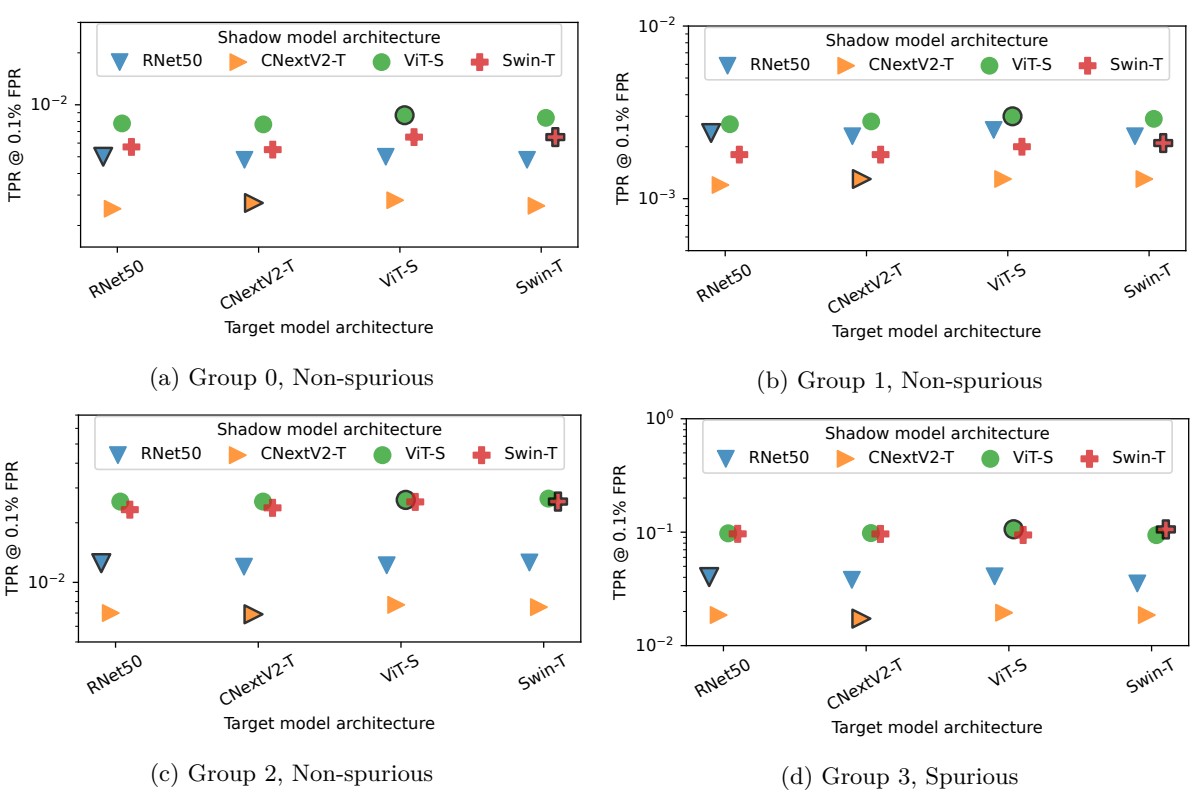

(a) Group 0, Non-spurious

(b) Group 1, Non-spurious

(c) Group 2, Non-spurious

(d) Group 3, Spurious

Figure 11: Varying the target and shadow model architecture per group on CelebA.

