# OpenReview forum: "Spurious Privacy Leakage in Neural Networks"
_TMLR — Accepted by TMLR_

### Review · Reviewer_p2bt · 2025-06-11

**Summary Of Contributions:**

This paper introduces the first work studying membership inference attacks (MIA) of DNNs trained on spurious correlated datasets. The authors show that across several datasets (Waterbirds, CelebA, FMoW, MultiNLI, and CivilComments), spurious groups exhibit much higher vulnerability to MIA attacks than other groups for lower complexity tasks (i.e. binary classification vs. multi-class). Additionally, the paper shows that existing mitigation methods against poor group performance (test acc. of spurious groups) improve performance but do not improve vulnerability to MIA, likely due to largely unchanged learned feature spaces. Finally, the authors show that different architectures beyond CNNs are also non-robust to datasets with spurious correlations.

**Audience:**

Yes

**Claims And Evidence:**

Yes

**Requested Changes:**

- [critical] the authors introduce evaluations on ViTs and other related architectures in section 5, but only evaluate them on general performance (worst group accuracy). Given the authors claim that ViTs do not appear more spurious robust than CNNs, it raises the question if MIA attacks against other architectures (other than CNNs) exhibit the same trend (spurious groups are more vulnerable). Can the authors add results for this to section 3?
- [strengthen] figure 1 does not appear to include FMoW results. I understand that there appears to be no spurious privacy leakage found, but I think it may still be useful to include in figure 1 for breadth.
- [strengthen] can the authors explain what the bolded values in Table 1 correspond to? While it is inferable it’s not immediately obvious

**Strengths And Weaknesses:**

**Strengths**

- interesting and novel results on privacy leakage for models trained on spurious datasets
- extensive results on several datasets

**Weaknesses**

- missing MIA results for different architecture (see requested changes)

---

> ### Author Response · Authors · 2025-06-12
>
> We thank the reviewer for the encouraging and valuable feedback. Below, we address the comments:
>
> > [critical] ... Given the authors claim that ViTs do not appear more spurious robust than CNNs, it raises the question if MIA attacks against other architectures (other than CNNs) exhibit the same trend (spurious groups are more vulnerable). Can the authors add results for this to section 3?
>
> Good point. Indeed it is a natural question to ask if other architectures than CNNs are also subject to spurious privacy leakage. The results in the Appendix Figure 8, 10 confirm that the most spurious groups remain significantly more vulnerable than non spurious groups. Across architectures (e.g. Resnet, Convnext, ViT, Swin), the difference between the most and the least spurious group is around one order of magnitude under TPR at 0.1% FPR (see Figure 8, groups 2, 3 for Waterbirds and Figure 10, groups 3, 1 for CelebA). This suggests that architecture choice alone does not mitigate spurious privacy leakage. To highlight these results, we will add a brief paragraph at the end of Section 5.
>
> > [strengthen] figure 1 does not appear to include FMoW results. I understand that there appears to be no spurious privacy leakage found, but I think it may still be useful to include in figure 1 for breadth.
>
> We agree with the reviewer and will add the full curve in the Appendix due to space limitations. Moreover, to avoid redundancy in the main text, the FMoW results are already presented in Figure 2a, using the TPR at 0.1% FPR metric instead of the full curve.
>
> > [strengthen] can the authors explain what the bolded values in Table 1 correspond to? While it is inferable it’s not immediately obvious
>
> Thanks for the suggestion. This will be added to the caption: [...] bolded values represent the best training method for privacy mitigation.

---

### Review · Reviewer_iEGd · 2025-06-21

**Summary Of Contributions:**

The authors find that spurious groups in datasets (i.e., groups affected by spurious correlation) are far more vulnerable to membership inference attacks (MIA) than other groups. Correspondingly, whereas task complexity reduces (e.g., fewer target classes in the data-set), privacy leakage for spurious groups remains the same or becomes worse, whereas leakage for other groups reduces. Moreover, whereas performance difference among groups gets mitigated through mechanisms such as DRO, DFR, and DP, those mechanisms still fail to eradicate privacy difference among spurious groups. Additionally, they show that models like Vision Transformers (ViTs) aren't necessarily any more robust to spurious correlations than convolutional models in controlled experimental settings.

**Audience:**

Yes

**Claims And Evidence:**

Yes

**Requested Changes:**

See the weaknesses.

**Strengths And Weaknesses:**

Strengths:

1. Key claims are backed by experiments, and the results are clearly presented with logical progression.

2. The paper is well structured and accessible, making it easy to follow the reasoning and methodology.


Weaknesses:

1. The motivation for examining privacy disparities across spurious vs. non-spurious subgroups needs stronger justification. Why should privacy parity across subgroups matter in practice? The paper could benefit from clearer discussion of the stakes, e.g., ethical, technical, or otherwise.

2. While the observations are technically interesting, their broader significance is unclear. How might these findings guide the design of new defenses or attacks? Without this, the results risk being perceived as a series of empirical notes rather than contributions to methodology or theory.

3. Some of the key claims rest on relatively narrow sets of experiments. More comprehensive evaluation, including across datasets, model scales, and attack variants, would help solidify the findings.

---

> ### Author Response · Authors · 2025-06-25
>
> We thank the reviewer for the valuable comments.
>
> > The motivation for examining privacy disparities across spurious vs. non-spurious subgroups needs stronger justification [...]
>
> We provide further motivation to introduce spurious privacy leakage:
>
> Privacy research often assumes that attacks affect all samples in the training data equally. This is a particularly problematic assumption for spurious correlated data, which partitions the dataset into subgroups of majority and minority groups. This raises a natural question: is there a significant privacy risk difference between different subgroups? For example, consider a medical dataset where 'young/healthy' and 'old/sick' dominate, with few 'young/sick' cases. A model learning this spurious correlation may memorize the 'young/sick' patients, possibly making them more vulnerable to privacy attacks. This privacy inequality adds on top of the fairness problem: groups already suffering from poor performance can simultaneously face higher privacy risks, violating both performance and privacy equity principles. In such a scenario, how can an auditor detect these fairness and privacy issues related to data biases?
>
> We will integrate these motivations in the introduction.
>
> > While the observations are technically interesting, their broader significance is unclear. How might these findings guide the design of new defenses or attacks? Without this, the results risk being perceived as a series of empirical notes rather than contributions to methodology or theory.
>
> To highlight the significance of our work, we discuss a few future directions that can benefit from our results:
> - For the fairness community, in particular for domain sensitive applications, we show that spurious correlation carries over from performance fairness to the privacy disparity, creating further ethical and privacy concerns (see the first answer).
> - For the security community, our results show that neither methods from spurious community (i.e. spurious robust training) nor DP seem suitable to be used as a practical solution. These limitations open doors for new defense methods that take into account privacy disparity beyond the worst-group accuracy. On the attack side, privacy disparity also presents opportunities to design more sophisticated methods. Our work reveals that adversaries can achieve higher MIA success by targeting groups with high spurious correlation strength. This suggests the possibility of new attack strategies that first identify spurious patterns and then focus attacks on vulnerable subgroups. For example, an adversary can inject poisoned samples to strengthen a spurious correlation between an attribute (e.g. race of a person) and the label. Then the attacker can perform MIA on the targeted subgroup with a higher success rate.
> - Lastly, another possible direction is in auditing real-world spurious data without explicit spurious attribute labels. In these cases, one can still use label-free spurious robust methods [1] to detect spurious correlations and perform auditing of both performance and privacy fairness, preventing undesired deployment. This is possible thanks to our results on spurious privacy leakage in real-world data.
>
> > Some of the key claims rest on relatively narrow sets of experiments. More comprehensive evaluation, including across datasets, model scales, and attack variants, would help solidify the findings.
>
> We acknowledge that a broader evaluation would strengthen our claims. At the same time, we believe our current experimental scope already provides extensive results on several datasets, as also noted by another reviewer. To recap, our experiments utilize (i) 5 different datasets, 3 training methods including the state-of-the-art spurious robust methods, and up to 8 significantly different model architectures (ii) real world datasets with natural occurring biases from both vision and text domains, (iii) rigorous evaluation using LiRA attack and extensive hyperparameter search including learning rates, weight decays, training epochs, spurious robust hyperparameters, and privacy budgets, and (iv) different MIA attacks including LiRA online, offline, and TrajMIA in Appendix Table 4. We believe our current evaluation sufficiently supports __our core claims regarding spurious privacy leakage, the ineffectiveness of spurious robust methods, and the role of architecture choice__. We acknowledge the limitation on larger-scale models and consider it as an important future work.
>
> While we believe our evaluation provides strong evidence, we welcome _specific_ suggestions for additional experiments (i.e. specific dataset or architecture) that can strengthen our core claims.
>
> [1] Yang et al. Identifying Spurious Biases Early in Training through the Lens of Simplicity Bias

---

### Review · Reviewer_jtch · 2025-06-26

**Summary Of Contributions:**

This work examines spurious privacy attacks in neural networks (NNs), specifically focusing on the impact of spurious correlation bias on privacy vulnerability. The authors find that reducing spurious correlation using spurious-robust methods does not mitigate privacy leakage. In particular, the study highlights privacy disparity based on memorization, where mitigating spurious correlation fails to prevent the memorization of spurious data—and thus does not improve the model’s overall privacy level. Additionally, the paper compares the privacy implications across different model architectures trained on spurious data, offering valuable insight into architecture-level sensitivity.

**Audience:**

Yes

**Claims And Evidence:**

Yes

**Requested Changes:**

I believe this paper be accepted as-is, with the following minor revisions:
- “where the dataset is divided into |G| groups”
- “reporte” to “reported” in Subsection 4.2.

**Strengths And Weaknesses:**

The claims made are clearly and concretely supported through extensive experimental verification and solid, well-structured writing. The results are significant and relevant, and the topic will be of interest to the privacy and fairness communities.

---

> ### Author Response · Authors · 2025-06-26
>
> Thank you for the positive recommendation and feedback! We are grateful to hear your appreciation of our experimental setup, writing structure, and the significance of our work. The requested changes will be addressed.

---

### Decision · Action_Editor_9thG · 2025-08-16

**Recommendation:** Accept as is

**Audience:**

Yes

**Audience Explanation:**

The reviewers reached a consensus that the paper provides valuable insights for the privacy research community.

**Claims And Evidence:**

Yes

**Claims Explanation:**

This paper exposes spurious privacy leakage, a phenomenon in which spurious groups are significantly more vulnerable to privacy attacks than non-spurious groups. It further shows that methods designed for spurious robustness do not mitigate this leakage, and that architectural configurations can influence its severity. The evaluations are conducted across multiple datasets and models, demonstrating empirical evidence.